# Socioeconomic inequalities in healthcare utilisation in Indonesia: a comprehensive survey-based overview

Joko Mulyanto,[1,2] Dionne S Kringos,[1] Anton E Kunst[1]

¹Department of Public Health, Amsterdam UMC, University of Amsterdam, Amsterdam Public Health Research Institute, Amsterdam, The Netherlands
²Department of Public Health and Community Medicine, Faculty of Medicine, Universitas Jenderal Soedirman, Purwokerto, Indonesia

**Correspondence to**
Joko Mulyanto;
j.mulyanto@amsterdamumc.nl

## ABSTRACT

**Objective** Monitoring inequality in healthcare utilisation is essential to reduce persistent inequalities in health in lower-middle income countries. This study aimed to assess socioeconomic inequalities in the utilisation of primary care, secondary care and preventive care in Indonesia.

**Methods** A cross-sectional study was conducted using data from the 2014 Indonesia Family Life Survey with a total of 42 083 adult participants. Socioeconomic status (SES) was measured by educational level and income. Healthcare utilisation was measured in: (1) primary care, (2) outpatient in secondary care, (3) inpatient care and (4) cardiovascular-related preventive care. The magnitude of inequalities was measured using the relative index of inequality (RII).

**Results** Small educational inequalities were found for primary care utilisation (RII 1.13, 95% CI 1.01 to 1.26). Larger educational inequalities were found for outpatient secondary care (RII 10.35, 95% CI 8.11 to 13.22) and inpatient care (RII 2.78, 95% CI 2.32 to 3.32). The largest educational inequalities were found for preventive care, particularly regarding blood glucose tests (RII 30.31, 95% CI 26.13 to 35.15) and electrocardiography tests (RII 30.90, 95% CI 24.97 to 38.23). Compared with educational inequalities, income inequalities were larger for primary care (RII 1.68, 95% CI 1.52 to 1.85) and inpatient care (RII 3.11, 95% CI 2.63 to 3.66), but not for outpatient secondary care and preventive care.

**Conclusions** Socioeconomic inequalities in healthcare utilisation in Indonesia are particularly large in secondary and preventive care. Therefore, it is recommended to prioritise policies focused on improving timely, geographical and financial access to secondary and preventive care for lower SES groups.

## INTRODUCTION

Equal use of healthcare for equal need is essential to improve population health and is, therefore, an objective for most healthcare systems. Monitoring inequality in healthcare utilisation is essential to assess the performance of a healthcare system, and ultimately to reduce persistent inequalities in health.[1] To monitor inequality in use in these terms, healthcare utilisation should be adjusted for self-assessed health (SAH) or morbidities, as determinants of healthcare need.[2 3]

### Strengths and limitations of this study

► This study was based on a nationally representative survey with a high response rate and with measurements that matched established international standards.
► Few studies have investigated inequalities in healthcare utilisation in Indonesia.
► The measurement of healthcare need was limited to self-assessed health.
► The measurement of healthcare utilisation was based on self-reported data which might be subject to recall bias.

There is evidence of inequalities in healthcare utilisation in developed countries, despite universal healthcare coverage.[4] For example, both in Western and Eastern Europe, inequalities in healthcare utilisation exist for certain types of healthcare. In Eastern Europe, the rapid transition of the healthcare system since the late 1990s after the fall of Communism may have been conducive to large inequalities.[5 6]

Lower-middle income countries (LMICs) also experience inequalities in healthcare utilisation especially in secondary care, as shown by an international comparative study in Asia, Africa and Latin America,[7] and studies in China and India.[8 9] Significant inequalities in healthcare utilisation are also found in Thailand, despite universal healthcare coverage since 2005.[10] For several reasons, sizeable inequalities in healthcare utilisation may also exist in other LMICs. Many of LMICs are struggling to provide universal healthcare coverage, resulting in persisting financial barriers to access healthcare. Furthermore, inadequate supply and unequal geographical distribution of healthcare facilities cause greater barriers to the use of these facilities by people living in remote places and with limited resources. Moreover, large inequalities in the quality of healthcare that is received may result from poor

stewardship, low financial investments in the healthcare system and suboptimal quality of a broad range of healthcare services.[11][12]

Indonesia is an LMIC with a population of 262 million people who are distributed across ≥17 000 islands, and with diverse ethnic and religious backgrounds. Indonesia's healthcare system is a mixture of public and private healthcare delivery systems. The size and role of the private–commercial healthcare market have increased during the last decade. Total health expenditure in 2014 was 2.8% of the gross domestic product, of which 47% originated from out-of-pocket payments. Since 1999 the government has provided health insurance for the poor, and in 2014 it introduced the National Health Insurance (NHI) programme to remove financial barriers to access basic healthcare services for the entire population by 2019.[13]

Current policy to achieve equal access in healthcare in Indonesia is focusing on the expansion of the NHI programme.[14] However, over the years, progress towards universal health coverage has been uneven and iterative and consistently driven by domestic political interests as opposed to technical considerations.[15] The dominance of political interest is also reflected in the government evaluation of the NHI programme which emphasised the overall coverage (NHI membership) of the population and paid less attention to the issue of the actual access distribution such as inequality among various population groups.[16]

In terms of preventive care, communicable diseases are still the government's priority with the improvement of universal child immunisation as the main focus.[16] Until recently, Indonesia did not implement a systematic policy or programmes for the prevention of cardiovascular diseases or other main non-communicable diseases (NCDs).[13] Furthermore, the NHI programme put much emphasis on curative care, which makes the utilisation of preventive care likely depend more on personal resources than on collective efforts.[17]

Lack of information which comprehensively assess the current situation of inequalities in healthcare utilisation in Indonesia may contribute to the low attention of the government in this issue. During the last decade, only a few studies have investigated inequalities in healthcare utilisation in Indonesia. Previous studies focused on the inequalities in maternal and child-related healthcare and dental care.[18–21] A recent report from the WHO stated that large inequalities in maternal and child healthcare persist in Indonesia, in addition to geographical inequalities in the healthcare infrastructure, particularly between the different provinces.[22] A recent study showed wealth-related inequalities in Indonesia in the use of healthcare, particularly in secondary care. However, this study did not assess inequalities in relation to other SES indicators such as educational level, nor did it consider inequalities in preventive care utilisation.[23]

No studies have empirically assessed socioeconomic inequalities (in terms of both educational level and income) in general healthcare utilisation in Indonesia particularly for preventive care utilisation. The present study aimed to fill in this gap of evidence. Using a large-scale national interview survey, we aimed to provide a comprehensive overview of socioeconomic inequalities in the utilisation of primary care, secondary care and preventive care in Indonesia. Findings from this study would be particularly beneficial for policymakers to assess the progress of the current efforts to reduce inequalities and also for policy development to further address inequalities in healthcare utilisation in Indonesia.

## METHODS

### Study design and data sources

We conducted a cross-sectional study using data from the fifth wave of the Indonesia Family Life Survey (IFLS5) which was conducted in 2014 by the RAND Corporation (USA). The IFLS5 is a longitudinal survey which has been conducted since 1993 (IFLS1) and collected data from 13 selected Indonesian provinces to maximally capture the diversity in the socioeconomic and cultural background of the Indonesian population. These 13 provinces represented 83% of the Indonesian population. The IFLS used stratified random sampling based on province and rural/urban location. The sampling frame was randomly chosen from the list enumeration area (EA) of the National Socioeconomic Survey which was conducted by the National Bureau of Statistics in more than 60 000 households. Within each urban EA, 20 households were randomly selected while 30 households were selected from each rural EA. In total, 7730 households from 321 EAs in 13 provinces were sampled for IFLS. The detail on IFLS data and supporting documents such as the survey protocol and questionnaires are publicly accessible through RAND's website.[24] The IFLS5 was approved by the relevant ethical review committees in the USA and Indonesia.

In our study, we included 42 083 individuals aged 15 years or older who had complete data for all study variables (98.2% of the total sample). For the analysis of cardiovascular-related preventive care utilisation, we included 26 612 individuals aged 31 years or older, which is 89.9% of the total number of individuals aged 31 years or older in the sample (29 612 individuals) and 63.2% of the total all-age sample (42 083 individuals). The present study excluded respondents aged 31 years or older because the risk of cardiovascular diseases substantially increases only after the age of 30 years.

### Measurements

The individual's educational level and income were used as indicators of socioeconomic status (SES). Educational level was defined according to the International Standard Classification of Education 2011 issued by UNESCO. Based on the highest level completed by each individual, educational level was categorised into pre-primary,

primary, lower secondary, upper secondary and tertiary level.

The level of household consumption was used as a proxy of income. In developing countries, consumption is considered a valid direct measurement of income or household wealth.[25] It measured at household level counted food, non-food consumables, durable goods, spending on education and housing. These counts were aggregated and transformed into a monthly consumption, which was adjusted for household size to consider the economics of scale.

We also adjusted for geographical differences in purchasing parity, using Jakarta's poverty line as a reference. Income measurement for different areas was adjusted taking into account variations in the poverty line by province, as well as urban versus rural place of residence. Data on the poverty line were obtained from the Indonesian Central Bureau of Statistics.

Healthcare utilisation data as collected by the IFLS5 were used. We measured the utilisation of: (1) (outpatient) primary care, (2) outpatient secondary care, (3) total outpatient care, (4) inpatient care and (5) cardiovascular-related preventive care. Primary care included any visits to or visits by trained health personnel from a public primary care centre, private primary care clinic and/or private primary care physician practice. Outpatient secondary care included any visit to a public hospital outpatient care (polyclinics) and private hospital outpatient care. The IFLS5 questionnaire measured all outpatient care that was received during a 4-week reference period.

Inpatient care was defined as any use of inpatient care during the previous 12 months for medical purposes, irrespective of the length of hospital stay. This included any use of inpatient care at primary care level with inpatient facilities, at public hospitals or private hospitals. For preventive care utilisation, we focused on cardiovascular diseases-related preventive care because of the sizeable contribution of cardiovascular diseases to the overall disease burden in Indonesia.[26] The use of cardiovascular risk factor screening was measured, including blood pressure measurements, cholesterol tests, blood glucose tests and ECG tests during the previous 12 months.

As a proxy of healthcare need, SAH was used. SAH is regarded as a health status measurement applicable to different socioeconomic groups. Data on SAH measurement from the IFLS5 survey were used, in which SAH was measured by asking 'In general, how is your health?'; the four response categories were 'very healthy', 'somewhat healthy', 'somewhat unhealthy' and 'very unhealthy'.

### Data analysis

To describe variation in healthcare use among socioeconomic groups, while taking into account differences between these groups in the age and sex structure, we calculated standardised prevalence rate (SPR) for each type of healthcare utilisation by educational levels and income quintiles. SPR was calculated as the number of cases per 100 persons and was standardised by age and sex using the direct method, with the total survey population as the standard population. Next, the rate difference and the rate ratio were calculated based on the SPR of the two lowest SES groups combined and the two highest SES groups combined, respectively. These SES groups were combined to provide a more stable estimation of the rate difference and the rate ratio between the lower and higher SES groups, respectively.[27] It complements the relative index of inequality (RII), as the latter takes into account all SES groups separately.

The RII was used to estimate the magnitude of inequalities in healthcare utilisation in a more comprehensive way. The RII is a regression-based index that assesses the probability of healthcare use in relationship to the relative hierarchical position of every individual within the socioeconomic hierarchy. We assigned the fractional rank of the socioeconomic indicators (income and educational) as the main predictor in the logistic regression model (considering the binary outcome of outpatient and inpatient care utilisation). The RII was obtained from the value of OR from the fractional rank of the socioeconomic indicators. The regression model was adjusted for age, sex and healthcare need, by controlling for SAH in the final model. Details on how RII calculated can be found elsewhere.[28]

A higher RII indicates a stronger association between the hierarchical position and healthcare utilisation and implies a greater difference in utilisation between higher SES groups compared with lower SES groups. More specifically, RII=1 indicates equality, RII <1 indicates higher utilisation among lower SES and RII >1 indicates higher utilisation among higher SES. The RII was chosen because it is commonly used in epidemiological research and has relatively a straightforward interpretation for readers who have no economics background compared with other common inequality measurements such as concentration index.

To correct for attrition and oversampling, the study sample was weighted with individual weights provided by the IFLS5. We used IBM SPSS Statistics V.24 as a statistical package to analyse the data.

### Patient and public involvement

No patients were involved in this study. Members of the public were not directly involved in this study.

### RESULTS

The study sample included slightly more women respondents (51.6%) than men (table 1). Almost two-thirds of the respondents were aged 15–45 years. Men had a generally higher level of education as compared with women. Majority of the respondents rated their health status as 'somewhat healthy' (58.8%). A more detail description of SAH among different SES groups is displayed in online supplementary table 1. Primary care was the most frequently used type of healthcare, with 14.6% of the

**Table 1** Basic characteristics of the study population

| Variables | Total n | % | Male n | % | Female n | % |
|---|---|---|---|---|---|---|
| **Gender** | | | | | | |
| Male | 20 374 | 48.4 | – | – | – | – |
| Female | 21 709 | 51.6 | – | – | – | – |
| **Age group (in years)** | | | | | | |
| 15–30 | 12 471 | 29.6 | 6436 | 31.6 | 6035 | 27.8 |
| 31–45 | 14 049 | 33.4 | 6545 | 32.1 | 7503 | 34.6 |
| 46–60 | 10 280 | 24.4 | 4973 | 24.4 | 5306 | 24.4 |
| >60 | 5283 | 12.6 | 2419 | 11.9 | 2864 | 13.2 |
| **Education level** | | | | | | |
| Pre-primary | 9868 | 23.4 | 3977 | 19.5 | 5891 | 27.1 |
| Primary | 9993 | 23.7 | 4855 | 23.8 | 5138 | 23.7 |
| Lower secondary | 8082 | 19.2 | 4041 | 19.8 | 4041 | 18.6 |
| Upper secondary | 10 731 | 25.2 | 5894 | 28.9 | 4838 | 22.3 |
| Tertiary | 3409 | 8.1 | 1607 | 7.9 | 1802 | 8.3 |
| **Income*** | | | | | | |
| 1st quintile (230–1300) | 8417 | 20.0 | 4050 | 19.9 | 4367 | 20.1 |
| 2nd quintile (1300–1830) | 8418 | 20.0 | 3997 | 19.6 | 4421 | 20.4 |
| 3rd quintile (1830–2520) | 8415 | 20.0 | 4056 | 19.9 | 4359 | 20.1 |
| 4th quintile (2520–3830) | 8417 | 20.0 | 4159 | 20.4 | 4258 | 19.6 |
| 5th quintile (3830–55 400) | 8416 | 20.0 | 4111 | 20.2 | 4305 | 19.8 |
| **Self-assessed health** | | | | | | |
| Very healthy | 8137 | 19.3 | 4362 | 21.4 | 3775 | 17.4 |
| Somewhat healthy | 24 757 | 58.8 | 12 179 | 59.8 | 12 578 | 57.9 |
| Somewhat unhealthy | 8447 | 20.1 | 3513 | 17.2 | 4934 | 22.7 |
| Very unhealthy | 742 | 1.8 | 320 | 1.6 | 422 | 1.9 |
| **Outpatient care utilisation** | | | | | | |
| Primary care | 6155 | 14.6 | 2006 | 9.8 | 4149 | 19.1 |
| Secondary care | 1022 | 2.4 | 427 | 2.1 | 595 | 2.7 |
| Total | 6864 | 16.3 | 2323 | 11.4 | 4541 | 20.9 |
| **Inpatient care utilisation** | | | | | | |
| Overall | 1937 | 4.6 | 591 | 2.9 | 1346 | 6.2 |
| **Preventive care utilisation (age≥31 years)** | | | | | | |
| Blood pressure screening | 21 663 | 80.5 | 9254 | 74.3 | 12 409 | 85.4 |
| Cholesterol screening | 4678 | 17.4 | 1951 | 15.7 | 2727 | 18.9 |
| Blood glucose screening | 4142 | 15.4 | 1855 | 14.9 | 2287 | 15.8 |
| ECG test | 1723 | 6.4 | 876 | 7.0 | 847 | 5.9 |

*Income in thousands Indonesian Rupiah.

respondents reporting that they utilised primary care at least once in the previous 4 weeks. The highest utilisation of preventive care was for blood pressure measurement, with 80.5% of the respondents reporting that their blood pressure was measured during the previous 12 months.

The prevalence rates of primary care use were about similar across all educational levels (table 2, see also online supplementary figure 1). Outpatient secondary care utilisation was more frequent among people with a higher educational level compared with people with a lower educational level. For overall inpatient care utilisation, the prevalence rates gradually increased with increasing educational level. A linear association was found between healthcare utilisation and income quintiles for all types

**Table 2** Standardised prevalence rate (SPR) of healthcare utilisation by socioeconomic status

| | Outpatient care (SPR, 95% CI)* | | | Inpatient care (SPR, 95% CI) |
| --- | --- | --- | --- | --- |
| | Primary | Secondary | Total | Overall |
| Education | | | | |
| Pre-primary | 14.47 (13.78 to 15.18) | 1.31 (1.12 to 1.53) | 15.22 (14.52 to 15.95) | 3.07 (2.77 to 3.40) |
| Primary | 14.93 (14.19 to 15.70) | 2.03 (1.76 to 2.32) | 16.07 (15.30 to 16.87) | 4.60 (4.18 to 5.04) |
| Lower secondary | 15.00 (14.12 to 15.91) | 1.88 (1.57 to 2.22) | 16.38 (15.47 to 17.33) | 4.31 (3.86 to 4.81) |
| Upper secondary | 14.38 (13.64 to 15.16) | 3.39 (3.03 to 3.79) | 16.99 (16.18 to 17.84) | 6.12 (5.63 to 6.64) |
| Tertiary | 14.37 (13.12 to 15.71) | 6.26 (5.44 to 7.17) | 18.97 (17.53 to 20.50) | 6.93 (6.04 to 7.91) |
| Income | | | | |
| 1st quintile | 11.71 (11.02 to 12.42) | 1.32 (1.09 to 1.57) | 12.54 (11.83 to 13.29) | 2.99 (2.65 to 3.35) |
| 2nd quintile | 14.11 (13.34 to 14.92) | 1.44 (1.20 to 1.71) | 14.73 (13.94 to 15.55) | 3.76 (3.37 to 4.18) |
| 3rd quintile | 15.36 (14.54 to 16.22) | 2.00 (1.71 to 2.33) | 16.75 (15.89 to 17.64) | 4.24 (3.81 to 4.70) |
| 4th quintile | 15.73 (14.88 to 16.62) | 2.91 (2.55 to 3.31) | 18.57 (17.64 to 19.53) | 5.13 (4.64 to 5.65) |
| 5th quintile | 16.13 (15.24 to 17.06) | 4.97 (4.48 to 5.50) | 19.83 (18.74 to 20.85) | 7.56 (6.94 to 8.21) |

*Prevalence rate per 100 persons, age and sex standardised to the total population.

of healthcare. This association was particularly strong for utilisation of outpatient secondary care and inpatient care.

Table 3 (see also online supplementary figures 2 and 3) quantifies the magnitude of educational and income inequalities in the utilisation of healthcare. Our findings from simple inequality measurement (rate ratio and rate difference) showed similarities with the findings from sophisticated inequality measurement (RII). No educational inequalities were found in primary care utilisation in the crude analysis, but positive educational inequalities (ie, higher education associated with higher use rates) emerged after adjusting for SAH (RII 1.13, 95% CI 1.01 to 1.26). We consistently found positive educational inequalities in all types and levels of healthcare use after adjusting for SAH. The largest educational inequality was

**Table 3** Socioeconomic inequalities in the utilisation of various types and levels of healthcare

| | Type of care | Level of care | SPR (95% CI)* | | Rate difference | Rate ratio | RII (95% CI), adjusted for age, sex | RII (95% CI), adjusted for age, sex, self-assessed health |
| --- | --- | --- | --- | --- | --- | --- | --- | --- |
| | | | Two lowest groups | Two highest groups | | | | |
| Education | Outpatient | Primary | 14.68 (14.18 to 15.20) | 14.38 (13.74 to 15.05) | −0.30 | 0.98 | 0.99 (0.98 to 1.01) | 1.13 (1.01 to 1.26) |
| | | Secondary | 1.64 (1.47 to 1.82) | 2.88 (2.59 to 3.19) | 1.24 | 1.76 | 7.89 (6.33 to 9.85) | 10.35 (8.11 to 13.22) |
| | | Total | 15.62 (15.09 to 16.15) | 17.51 (16.80 to 18.25) | 1.89 | 1.12 | 1.35 (1.24 to 1.46) | 1.59 (1.44 to 1.77) |
| | Inpatient | Overall | 3.74 (3.48 to 4.01) | 6.31 (5.88 to 6.76) | 2.57 | 1.69 | 2.38 (1.97 to 2.76) | 2.78 (2.32 to 3.32) |
| Income | Outpatient | Primary | 12.88 (12.35 to 13.42) | 16.25 (15.62 to 16.89) | 3.37 | 1.26 | 1.50 (1.39 to 1.62) | 1.68 (1.52 to 1.85) |
| | | Secondary | 1.36 (1.20 to 1.54) | 4.69 (4.35 to 5.04) | 3.33 | 3.45 | 6.61 (5.29 to 8.25) | 7.43 (5.88 to 9.39) |
| | | Total | 13.61 (13.08 to 14.16) | 19.18 (18.50 to 19.88) | 5.57 | 1.41 | 1.80 (1.67 to 1.94) | 2.15 (1.96 to 2.36) |
| | Inpatient | Overall | 3.36 (3.10 to 3.63) | 6.30 (5.91 to 6.71) | 2.94 | 1.88 | 2.94 (2.52 to 3.43) | 3.11 (2.63 to 3.66) |

*Prevalence rate per 100 persons, age and sex standardised to the total population.
SPR, standardised prevalence rate; RII, relative index of inequality.

**Table 4** Standardised prevalence rate (SPR) of preventive care utilisation by socioeconomic status

| | Preventive care activity (SPR, 95% CI)* | | | |
| --- | --- | --- | --- | --- |
| | **Blood pressure** | **Cholesterol** | **Blood glucose** | **ECG** |
| Education | | | | |
| Pre-primary | 72.40 (70.54 to 74.30) | 8.85 (8.24 to 9.49) | 6.85 (6.32 to 7.41) | 2.51 (2.19 to 2.86) |
| Primary | 79.54 (77.47 to 81.65) | 10.78 (10.03 to 11.58) | 10.59 (9.84 to 11.39) | 4.00 (3.54 to 4.50) |
| Lower secondary | 82.72 (79.88 to 85.63) | 18.03 (16.67 to 19.48) | 15.44 (14.17 to 16.79) | 4.95 (4.25 to 5.73) |
| Upper secondary | 86.66 (84.27 to 89.10) | 26.83 (25.44 to 28.27) | 24.72 (23.39 to 26.10) | 10.80 (9.94 to 11.71) |
| Tertiary | 92.33 (88.45 to 96.35) | 44.72 (41.97 to 47.60) | 43.38 (40.66 to 46.23) | 20.99 (19.13 to 22.99) |
| Income | | | | |
| 1st quintile | 73.55 (71.27 to 75.88) | 7.83 (7.11 to 8.60) | 6.12 (5.49 to 6.81) | 2.71 (2.29 to 3.18) |
| 2nd quintile | 76.77 (74.45 to 79.15) | 12.12 (11.21 to 13.08) | 10.14 (9.30 to 11.02) | 3.37 (2.90 to 3.90) |
| 3rd quintile | 80.60 (78.22 to 83.03) | 14.37 (13.37 to 15.42) | 12.63 (11.70 to 13.63) | 4.64 (4.08 to 5.26) |
| 4th quintile | 83.68 (81.25 to 86.16) | 20.34 (19.14 to 21.59) | 18.41 (17.27 to 19.60) | 7.13 (6.43 to 7.89) |
| 5th quintile | 87.87 (85.38 to 90.41) | 32.69 (31.17 to 34.26) | 30.09 (28.64 to 31.60) | 14.43 (13.43 to 15.49) |

*Prevalence rate per 100 persons, age and sex standardised to the total population.

found in outpatient secondary care utilisation (RII 10.35, 95% CI 8.11 to 13.22).

Positive income inequalities (ie, higher income associated with higher use rates) were found in all types and levels of healthcare use, especially after adjustment for SAH. Similar to educational inequalities, the largest income inequality was found in outpatient secondary care utilisation (RII 7.43, 95% CI 5.88 to 9.39). Generally, larger inequalities were found in relationship to income as compared with educational level, except for utilisation of outpatient secondary care.

A consistent linear association was found between prevalence rate of preventive care utilisation and SES (table 4, and online supplementary figure 1). The prevalence rate of blood pressure measurement increased incrementally by SES group for both educational level and income quintiles. The prevalence rate of cholesterol tests, blood glucose tests and ECG tests drastically increased from the third highest SES groups to the highest SES groups, both for income and educational level. The differences were larger in relationship to educational level than to income.

Table 5 shows the estimates of the size socioeconomic inequalities in preventive care utilisation (see also online supplementary figures 2 and 3). Our analyses showed consistent findings between simple (rate difference and rate ratio) and sophisticated inequality estimations (RII). Exceptionally large positive educational inequalities were found in blood glucose tests (RII 30.31, 95% CI 26.13 to 35.15) and ECG tests (RII 30.90, 95% CI 24.97 to 38.23). For income inequalities, inequalities in preventive care utilisation were smaller compared with educational inequalities. ECG tests showed the largest income inequality (RII 12.96, 95% CI 10.68 to 15.73), and blood pressure measurements showed the smallest income inequality (RII 3.40, 95% CI 3.04 to 3.79).

## DISCUSSION

This study documented socioeconomic inequalities in healthcare utilisation among the adult population in Indonesia. These inequalities were particularly large for secondary and preventive care. Compared with educational inequalities, income-related inequalities were larger for primary care and inpatient care but smaller for outpatient secondary and preventive care.

This study was based on a nationally representative survey with a high response rate (95.3%) and with measurements that matched established international standards.[29] A possible limitation of the study is the measurement of healthcare need that was limited to SAH. Ideally, we would have used multiple measures of healthcare need such as self-reported morbidities or health functioning. Although our dataset provided self-reported morbidities and data on health functioning, these are likely to be underestimated in the Indonesian population (particularly in lower SES groups),[30] and therefore invalid for healthcare need adjustments.

Because no registry-based data on inequalities in healthcare utilisation in Indonesia are available, we used self-reported use of healthcare. Such healthcare utilisation measures may be subject to recall bias. However, the problem of recall bias might be limited, as the prevalence values of outpatient and inpatient care utilisation from the IFLS5 are close to the national average in Indonesia as reported by the Ministry of Health and data from the National Economic Survey.[31 32]

Previous studies in Indonesia mostly focus on specific healthcare services such as maternal and child-related healthcare. Our findings show that the direction and magnitude of inequalities in healthcare use among individuals aged 15 years or older bear a resemblance to the large socioeconomic inequalities in maternal healthcare

**Table 5** Socioeconomic inequalities in the use of various preventive care activities

|  | Activity | SPR (95% CI)* | | Rate difference | Rate ratio | RII (95% CI) adjusted for age, sex | RII (95% CI) adjusted for age, sex, SAH |
|---|---|---|---|---|---|---|---|
|  |  | Two lowest groups | Two highest groups |  |  |  |  |
| Education | Blood pressure | 75.77 (74.38 to 77.17) | 88.28 (86.24 to 90.36) | 12.51 | 1.17 | 6.37 (5.61 to 7.24) | 6.67 (5.87 to 7.59) |
|  | Cholesterol | 9.69 (9.22 to 10.19) | 32.12 (30.85 to 33.44) | 22.43 | 3.31 | 18.17 (15.91 to 20.74) | 21.27 (18.53 to 24.41) |
|  | Blood glucose | 8.47 (8.03 to 8.94) | 30.21 (28.98 to 31.49) | 21.74 | 3.57 | 24.61 (21.36 to 28.36) | 30.31 (26.13 to 35.15) |
|  | ECG | 3.17 (2.90 to 3.46) | 13.74 (12.92 to 14.60) | 10.57 | 4.33 | 25.45 (20.72 to 31.20) | 30.90 (24.97 to 38.23) |
| Income | Blood pressure | 75.17 (73.54 to 76.83) | 86.28 (84.54 to 88.05) | 11.11 | 1.15 | 3.34 (2.99 to 3.72) | 3.40 (3.04 to 3.79) |
|  | Cholesterol | 8.61 (8.06 to 9.17) | 26.54 (25.57 to 27.53) | 17.93 | 3.08 | 9.20 (8.15 to 10.40) | 9.76 (8.63 to 11.02) |
|  | Blood glucose | 8.11 (7.59 to 8.66) | 24.29 (23.36 to 25.54) | 16.18 | 3.00 | 10.81 (9.49 to 12.30) | 11.59 (10.18 to 13.20) |
|  | ECG | 3.03 (2.71 to 3.37) | 10.78 (10.14 to 11.40) | 7.73 | 3.55 | 12.42 (10.23 to 15.07) | 12.96 (10.68 to 15.73) |

*Prevalence rate per 100 persons, age and sex standardised to the total population.
SAH, self-assessed health; SPR, standardised prevalence rate; RII, r relative index of inequality.

and child healthcare.[18 22] Similar to the recent study on wealth-related inequality in healthcare utilisation in Indonesia, we found smaller inequalities in the utilisation of primary care, especially outpatient care, and larger inequalities in secondary care.[23] Our results are also consistent with studies performed in other LMICs showing relatively small inequalities in PC utilisation and larger inequalities in secondary care.[7–10]

The small socioeconomic inequalities in primary care utilisation are probably related to the relatively high supply and geographical distribution of primary care providers in Indonesia. Of all registered physicians in Indonesia, 78.4% are general practitioners who mostly practice as public or private providers. In total, there are 9745 public primary care centres providing services for the national population with subsidy by local governments.[13 33] Moreover, according to recent studies, access to primary care was increased by a government-financed NHI programme that aimed to reduce financial barriers of the poor population to healthcare.[23 32 34] In the NHI programme, primary care acted as gatekeeper which required all the beneficiaries regardless of their socioeconomic background (poor people or government employee) to use primary care as an entry point to access the healthcare service.[35] For people without insurance coverage, primary care is relatively affordable and can be accessed at low cost even in private practices.[13] This likely explained the smaller income and education-related inequalities in the primary care utilisation compared with the inequalities in secondary and inpatient care utilisation.

In contrast to primary care, the use of secondary care facilities in Indonesia showed considerable inequalities by both educational level and income. For example, individuals with the highest income had seven times higher odds to use outpatient secondary care compared with those with the lowest income. It is likely that geographical barriers contribute to these inequalities. Because most secondary care facilities and specialists are located in urban areas, the poor people need to pay high indirect costs (in terms of travel and opportunity) to access secondary care, even if their medical costs are covered by the NHI programme.[23 32 34] Moreover, there is a limited supply of secondary care specialists. These specialists tend to work as private for-profit healthcare providers, who are not contracted by the NHI programme. This is likely to result in low financial access for lower SES groups rather than higher SES groups, which may have supplementary private health insurance.[18 21]

We observed inequalities to be larger outpatient secondary care than for inpatient care. A possible explanation is that outpatient secondary care is much more affordable for higher income groups than for lower groups, as the former can pay the service by out-of pocket payment or private health insurance. Lower income groups generally can use outpatient secondary care only by using government health insurance with its referral system. For inpatient care, however, utilisation costs are significant for higher income groups as well as lower income groups and usually

only affordable via government health insurance and accessible through a referral system.[21]

Inefficient referral procedures could also have contributed to larger inequalities in secondary care utilisation compared with primary care, particularly for education-related inequalities. Even when low-educated people are entitled to access secondary healthcare, they may lack the knowledge required to obtain a referral due to the complexity of the administrative procedures in the referral system.[34] Inequalities in secondary care may also be influenced by differences between educational groups in the preferences and resources that influence the way people use healthcare.[36] An Indonesian study showed that patients with higher educational level, regardless of their income level, were more likely to judge the quality of primary care to be low and to ask for a referral to secondary care. This tendency was not observed among people with high income but relatively low education.[37 38] Education-related preferences might explain why educational inequalities in outpatient secondary care were larger compared with income-related inequalities.

We observed exceptionally large socioeconomic inequalities in preventive care, particularly by education. For example, individuals who had the highest educational level had 30 times higher odds to have a blood glucose test in the previous 12 months compared with those who had the lowest educational level. The individuals' level of health literacy may play a major role in their use of preventive care.[39] Those with a relatively low level of health literacy may experience cognitive barriers to make decisions regarding diagnostic tests and treatments that they may need, irrespective of financial, geographical or administrative barriers.[40 41] It also likely explains relatively smaller education-related inequalities in blood pressure measurement compared with other types of preventive care because blood pressure disorder such as high blood pressure is relatively known by common people regardless of their educational background compared with other types of preventive care.

The exceptionally large inequalities in preventive care utilisation may reflect the low priority given to preventive care in Indonesia's health policy which to date has strongly focused on curative care. This resulted in low health expenditures on preventive care,[42] and the absence of a nationwide preventive programme for the NCDs. As a result, the utilisation of preventive care is relying more on personal resources or potentially motivated or initiated by physicians who have more attention to preventive care.[43 44]

## CONCLUSIONS

The findings underline the need to develop comprehensive efforts to tackle significant socioeconomic inequalities in healthcare utilisation in Indonesia. Potential areas of priority include removing financial and geographical barriers by providing the NHI programme with universal health coverage, improving the supply and distribution

of secondary care services, simplifying the referral system procedure and developing a nationwide preventive care programme. Improving the quality of primary care by providing better infrastructure and developing the competence of health personnel may have large impact on population health considering the (equality in) accessibility of primary care and could potentially reduce the burden of secondary care. Monitoring healthcare (in) equality will be essential to evaluate the impact of these policies. Further research is needed to assess inequalities in healthcare among specific patient groups, to evaluate the contribution of patient preferences and resources and to examine the role of geographical factors and healthcare organisation and infrastructure. Such in-depth analyses could provide a better understanding of socioeconomic inequalities in healthcare utilisation in Indonesia and guide the development of strategies to address those inequalities.

**Acknowledgements** The authors thank the RAND Corporation for providing the dataset of the Indonesian Family Life Survey (IFLS) for this study.

**Contributors** JM conceived the paper. JM and AEK developed the analysis strategy. JM conducted the data analysis. JM, DSK and AEK collectively interpreted the findings. JM prepared the initial draft of the manuscript. JM, DSK and AEK equally contributed to the revision of the manuscript. All authors have read and approved the final manuscript.

**Funding** This study is funded by the Indonesian Endowment Fund for Education (LPDP), Ministry of Finance, Republic of Indonesia, grant number 20160322045795. The funding source has no role in study design, data collection, data analysis and interpretation, manuscript writing and decision to submit or publish.

**Competing interests** None declared.

**Patient consent for publication** Not required.

**Ethics approval** This study is a secondary analysis using the Indonesian Family Life Survey (IFLS) dataset. The IFLS was approved by the Institutional Review Board (IRB) of the Rand Corporation (USA) and the Survey Meter (Indonesia). The data set is publicly available and no personal information can be identified. This study is categorised as being exempt from human research according to the National Institute of Health (NIH).

**Provenance and peer review** Not commissioned; externally peer reviewed.

**Data sharing statement** This study used the Indonesian Family Life Survey (IFLS) dataset provided by RAND Corporation. The IFLS dataset is freely accessible at https://www.rand.org/labor/FLS/IFLS.html. Additional unpublished data are available by request to the corresponding author.

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
