## [Reviewer comments · BMJ Open]

ARTICLE DETAILS

TITLE (PROVISIONAL)	Socioeconomic inequalities in healthcare utilisation in Indonesia: a comprehensive survey-based overview
AUTHORS	Mulyanto, Joko; Kringos, Dionne Sofia; Kunst, Anton

VERSION 1 – REVIEW

REVIEWER	Shiow-Ing Wang Asia University, Taiwan
REVIEW RETURNED	03-Sep-2018

GENERAL COMMENTS	1. Description trend :You should used the data of 1st ~5th wave of the Indonesia Family Life Survey (IFLS) to describe the trend of inequality in healthcare utilization. 2. Precise definition:"no studies have empirically assessed socioeconomic inequalities in general healthcare utilization in the adult population in Indonesia", 15 years old are not adult. 3.Calculation error :26,612 individuals aged ≥31 years (99.1% of the total subsample).$26612/42083=63.23\%$ 4. Description error : " Utilisation of cardiovascular-related preventive care... individuals aged ≥31 years ; the present study excluded respondents aged ≤31 years ...", Which category does the 31-year-old case belong to? 5. Methodology is not detailed enough: for example, cardiovascular risk factor screen, who are the target of screening? How to implement? How to do it? Need a doctor referral?This will affect the conclusion of the study. 6. Add value: you should provide some information to add the value of study. such as the distribution of SAH, utilization of healthcare among different SES groups. Table 1 you provide the information between gender group, but the study is focus on SES. 7. Conclusion should be cautious: Whether the electrocardiography tests and blood glucose tests need to be tested should be judged by the doctor, not the education level of the patient. You should explore the hidden reasons. "Potential areas of priority include removing financial and geographical barriers....", this study does not provide information on geographic barriers. 8. New findings or contributions: When you are modifying this article, please focus on "What new findings did this research provide?", Because there are too many papers have told us that education and income level will affect the equity of healthcare utilization.
--

REVIEWER	Henry Wilde Chulalongkorn University. Bagkok Thailand
REVIEW RETURNED	18-Sep-2018

GENERAL COMMENTS	This is an interesting subject and well done and professionally
---

	reported study that provides virtually all the information needed to form an opinion concerning this topic. The data collection, numbers of subjects and their selection, statistical methods, selection of analysts appear to be of sufficient quality for the tasks. From my perspective as an infectious disease doctor and clinical investigator working mostly in southeast Asia here are the conclusion that I would offer:  1) The work, exclusively from continental and the islands Indonesia, was very challenging and results are not surprising to me as being well familiar to me for decades. 2) Health care still needs much improvement in material, staffing and funding for patent care. 3) The quality of Primary Care is deficient in all parameters except perhaps numbers of providers. 4) Secondary and higher-level care has not yet been well defined, quantitated and is also not fully utilized. 5) Preventive medicine is still in its infancy in respect to universal immunization availability, preventive medicine education re smoking avoidance, dietary modifications and related efforts. 6) Not much information can be analyzed regarding infectious disease prevention or early detection and control 7) Indonesia has huge barriers to development due to the fact that it is a huge country composed of thousands of isolated islands. 8) Providing quality primary care and costly secondary will still require funding, infrastructure building and education of primary care staff as well as innovative secondary specialists that can build a suitable health care system that will have to be innovative and require much support.
--	---

REVIEWER	Zhonghua Wang Nanjing medical university, China
REVIEW RETURNED	22-Jan-2019

GENERAL COMMENTS	The reviewed manuscript evaluated socioeconomic inequalities in healthcare utilisation in Indonesia using the data from a national representative survey. Educational inequalities and income inequalities in different types and levels of health care use were measured by the relative index of inequality(RII). The findings have potential to help guide future policy decisions. The manuscript has some general and specific concerns.  1. method  a. Page6. Study design and data sources. Additional information is required about the IFLS5 survey methodology so that the reader would probably benefit from a direct reference to the IFLS5 baseline protocol or methodology paper, to see how the sample was selected and how it is representative of the population. b. Page 8 line 24. I recommend that the author gives a more detailed introduction about RII. For example, the rationale for the analysis should be well explained, allowing for the study to be repeated. 2. result  a. In the data analysis, the author mentioned “the rate difference and the rate ratio complement the RII”. Thus, i think the rate difference and the rate ratio were also important ways of describing inequalities of healthcare utilisation in the manuscript. Additional information is required about the findings of it in the result section, like value and meaning. 3. discussion  a. In the result section, the author mentioned “ larger inequalities were found in relationship to income as compared to educational
---

	level, except for utilisation of outpatient secondary care.”. I think the author should further explain the reasons in the discussion section. For example, why the educational inequalities were larger than income inequalities for outpatient secondary care? b. There is no plan for the next step in this manuscript.
--	---

REVIEWER	Mohammad Habibullah Pulok Postdoctoral Research Associate Nova Scotia Health Authority, Halifax, Canada.
REVIEW RETURNED	30-Jan-2019

GENERAL COMMENTS	Thank you very much for giving me the opportunity to review a nicely articulated paper. The article on socioeconomic inequalities in healthcare utilisation in Indonesia is an interesting and well-designed study. I do not have any major comments that could add something important to the manuscript. The paper is well-written, and the methodology of the paper is sound and robust. The authors have applied the method very well to the available data. Minor comments: I would suggest the authors develop the motivation of the paper a little bit. The introduction would be more attractive if the authors could draw a storyline at the end of the introduction. The motivation is currently statistical which lacks an argument to conduct the study. In this case, the authors could consider adding the current policy perspective in Indonesia to achieve equity. The authors could improve the discussion section by explaining the possible reasons for lower education-related inequality in some outcomes of healthcare use. The author could also add one or two sentences to outline the future direction of research in the Indonesian context. I would suggest the authors to consider the following recent paper from Indonesia to get some insights to enrich the manuscript. Johar, M., Soewondo, P., Pujisubekti, R., Satrio, H.K. & Adji, A. 2018, 'Inequality in access to health care, health insurance and the role of supply factors', Social Science & Medicine, vol. 213, pp. 134–45
---

VERSION 1 – AUTHOR RESPONSE

Reviewer 1

Reviewer Name: Shioh-Ing Wang

Institution and Country: Asia University, Taiwan

We thank for the valuable comments and constructive feedback. We provide our response to the specific comments (in italic), as follow:

1. Description trend: You should use the data of 1st ~5th wave of the Indonesia Family Life Survey (IFLS) to describe the trend of inequality in healthcare utilization.

Reply: We understand the value of trend analysis, but in this paper, we would prefer to focus on the analysis of the current situation of socioeconomic inequalities in healthcare utilisation in Indonesia using the most recent available data, as we mentioned in our objective (introduction section, paragraph 9, sentence 3). The inclusion of trends analysis in the current paper would give too much information for one single paper; it would be a paper on its own.

2. Precise definition: "no studies have empirically assessed socioeconomic inequalities in general healthcare utilization in the adult population in Indonesia", 15 years old are not adult.

Reply: Point well taken. Our definition of an adult individual referred to the definition used by IFLS dataset in their sampling method which included all individuals aged 15 years or older. Following your suggestion, we have deleted the word "adult" and revised our sentence to make it more precise, as follows:

Introduction section, paragraph 9, sentence 1:

"No studies have empirically assessed socioeconomic inequalities (in terms of both educational level and income) in general healthcare utilisation in Indonesia, particularly for preventive care utilisation. The present study aimed to fill in this gap of evidence."

Discussion section, paragraph 4, sentence 1-2:

"Previous studies in Indonesia mostly focus on specific healthcare services such as maternal and child-related healthcare. Our findings show that the direction and magnitude of inequalities in healthcare use among individuals aged 15 years or older bear resemblance to the large socioeconomic inequalities in maternal healthcare and child healthcare.16 20"

3. Calculation error: 26,612 individuals aged ≥ 31 years (99.1% of the total subsample).
 $26612/42083=63.23\%$

Reply: Thank you for the correction. What we intended to do was to describe the proportion of individuals aged 31 years or older included in the analysis compared to the total number of individuals aged 31 years or older in the sample. This makes 26,612 out of 29,612=89.9%. We have revised the sentences to make it clearer, as follows:

Methods section, subheading study design and data source, paragraph 1, sentence 6:

"For the analysis of cardiovascular-related preventive care utilisation, we included 26,612 individuals aged 31 years or older, which is 89.9% of the total number of individuals aged 31 years or older in the sample (29,612 individuals) and 63.2% of the total all-age sample (42,083 individuals)"

4. Description error: " Utilisation of cardiovascular-related preventive care... individuals aged ≥ 31 years; the present study excluded respondents aged ≤ 31 years ...", Which category does the 31-year-old case belong to

Reply: We have revised the sentence, as follows:

Methods section, subheading study design and data sources, paragraph 1, sentence 7:

"The present study excluded respondents aged less than 31 years because the risk of cardiovascular diseases substantially increases only after the age of 30 years."

5. Methodology is not detailed enough: for example, cardiovascular risk factor screen, who is the target of screening? How to implement? How to do it? Need a doctor referral? This will affect the conclusion of the study.

Reply: The proposed level of detail would be beyond the scope of our paper. Moreover, the IFLS dataset that we used did not contain any information regarding this issue. However, we understand that some more background information on the (lack of) focus on preventive care and screening in Indonesia is important to provide a better interpretation of our findings. We have therefore added some text on this issue in the introduction section, and later in the discussion section, as follows:

Introduction section, paragraph 6:

“In terms of preventive care, communicable diseases are still the government’s priority with the improvement of universal child immunisation as the main focus.¹⁶ Until recently, Indonesia did not implement a systematic policy or program for the prevention of cardiovascular diseases or other main non-communicable diseases (NCDs).¹³ Furthermore, the NHI program put much emphasis on curative care, which makes the utilisation of preventive care likely depend more on personal resources than on collective efforts.¹⁷”

Discussion section, paragraph 10:

“The exceptionally large inequalities in preventive care utilisation may reflect the low priority given to preventive care in Indonesia’s health policy which to date has strongly focused on curative care. This resulted in low health expenditures on preventive care, and the absence of nationwide programs for the prevention of NCDs. As a result, the utilisation of preventive care is relying more on personal resources or potentially motivated or initiated by physicians who have more attention to preventive care.”

6. Add value: you should provide some information to add the value of the study, such as the distribution of SAH, utilization of healthcare among different SES groups. Table 1 you provide the information between gender group, but the study focuses on SES.

Reply: Due to the limitation of space with regards to journal’s requirement, and considering the amount of information which we have to display to address our objectives; we decided not to display the requested information in the main results section. Instead, we did add a new table as a supplementary file to display this information (see the supplementary file, table 1).

For distribution of healthcare utilisation among different SES groups, we have comprehensively displayed relevant information in table 2 and table 4, and in the supplementary file figure 1 and figure 2.

7. Conclusion should be cautious:

a. Whether the electrocardiography tests and blood glucose tests need to be tested should be judged by the doctor, not the education level of the patient. You should explore the hidden reasons.

Reply:

We agree that ideally the use of electrocardiography and blood glucose test should be based on the physician’s diagnosis. However, in the absence of a nation-wide systematic preventive program, health knowledge and awareness, which are associated with a patient’s educational level, is likely influencing individual’s decision to use preventive care (see discussion section, paragraph 9). In the discussion section, we discuss possible reasons for this influence of educational level.

b. "Potential areas of priority include removing financial and geographical barriers....", this study does not provide information on geographic barriers.

Reply

We acknowledged that our study did not provide any direct results regarding geographical barriers. However, we have provided several plausible explanations (discussion section, paragraph 6 and 8) which would imply a role of geographical aspects of healthcare (such as unequal distribution of health personnel). Several recent studies in Indonesia showed that geographical barriers might play an important role in the inequalities in healthcare utilisation in Indonesia. However, we agree that there is a lack of evidence from our own study. Therefore, we phrased our conclusion carefully using the word “potential”, and we emphasised the need for further exploration. We added several sentences to the conclusion section, paragraph 1, sentence 6-7, as follows:

“Further research is needed to assess inequalities in healthcare among specific patient groups, and to evaluate the contribution of patient preferences and resources, and to examine the role of geographical factors and healthcare organisation and infrastructure. Such in-depth analyses could provide a better understanding of socioeconomic inequalities in healthcare utilisation in Indonesia and guide the development of strategies to address those inequalities.”

8. New findings or contributions: When you are modifying this article, please focus on "What new findings did this research provide?", Because there are too many papers, have told us that education and income level will affect the equity of healthcare utilization.

Reply: We have highlighted the added value of this paper in the introduction section (paragraph 9) and the discussion section (paragraph 4). This study provided the first comprehensive description in socioeconomic inequalities in healthcare inequalities in Indonesia that includes both curative and preventive care, and that considers two measures of SES (educational level and income). Even though our observations on inequalities are not surprising, our detailed description of these inequalities provides information needed for further development of such policies in Indonesia. We have added the following sentences to make the added value of our study clearer for the readers:

Introduction section, paragraph 5, sentence 3:

“The dominance of political interest is also reflected in the government monitoring and evaluation of the NHI program which emphasised the overall coverage (NHI membership) of the population and paid less attention to the issue of the actual access distribution such as inequality among various population groups.”

Introduction section, paragraph 8, sentence 1-3:

“Lack of information which comprehensively assesses the current situation of inequalities in healthcare utilisation in Indonesia may contribute to the low attention of the government to this issue. During the last decade, only a few studies have investigated inequalities in healthcare utilisation in Indonesia. Previous studies focused on inequalities in maternal and child-related healthcare and dental care.”

Introduction section, paragraph 9, sentence 4:

“Findings from this study would be particularly beneficial for policymakers to assess the progress of the current efforts to reduce inequalities and also for policy development to further address inequalities in healthcare utilisation in Indonesia.”

Reviewer: 2

Reviewer Name: Henry Wilde

Institution and Country: Chulalongkorn University. Bangkok Thailand

Comments to Authors

This is an interesting subject and well done and professionally reported study that provides virtually all the information needed to form an opinion concerning this topic. The data collection, numbers of subjects and their selection, statistical methods, selection of analysts appear to be of sufficient quality for the tasks. From my perspective as an infectious disease doctor and clinical investigator working mostly in southeast Asia here is the conclusion that I would offer:

Thank for the thorough review of our study. We provide our response to the specific comments (in italic), as follow:

1. The work, exclusively from continental and the islands Indonesia, was very challenging and results are not surprising to me as being well familiar to me for decades.

Reply: We agree that in general our results were not surprising and similar to the results from previous studies regarding inequalities in specific healthcare utilisation such as maternal and child-

related healthcare. We have highlighted the added value in our response to comment 8 of reviewer 1. We would like to refer to this response.

2. Health care still needs much improvement in material, staffing and funding for patient care.

Reply: We very much agree with the comment. In fact, our findings regarding outpatient secondary care and inpatient care supported this statement. We mention this in our discussion section paragraph 5 and 6.

3. The quality of Primary care is deficient in all parameters except perhaps numbers of providers.

Reply: Our study did not study/provide information about the quality of primary care. Therefore, we were not able to discuss this issue in depth. However, we agree that Indonesia has a relatively large number of primary care providers as we mentioned in our discussion section, paragraph 5.

4. Secondary and higher-level care has not yet been well defined, quantitated and is also not fully utilized.

Reply: We agree that the supply of secondary and high-level care is not adequate and there are also technical problems that may hamper the utilisation of health care such as the referral procedure. We have discussed this issue in our discussion section paragraph 6-7.

5. Preventive medicine is still in its infancy in respect to universal immunization availability, preventive medicine education re smoking avoidance, dietary modifications and related efforts.

Reply: Our study did not provide any information regarding universal immunization, smoking avoidance, dietary modifications, and other primary prevention efforts. Therefore, we were not able to further address those issues in our discussion. However, we agree that in general, preventive care until recently is not a priority in Indonesia's context. We mentioned this in the introduction section, paragraph 7 and the discussion section, paragraph 10.

6. Not much information can be analyzed regarding infectious disease prevention or early detection and control.

Reply: We agree that infectious disease and their prevention is still an important issue in Indonesia. Unfortunately, the dataset which was used in our study did not provide any data regarding infectious diseases prevention. Therefore, we were unable to carry out further analysis regarding this issue in our study.

7. Indonesia has huge barriers to development due to the fact that it is a huge country composed of thousands of isolated islands.

Reply: We agree that geographical barriers may play an important role in the access of healthcare in Indonesia particularly for secondary care. We discussed this issue in the discussion section, paragraph 7-8.

8. Providing quality primary care and costly secondary will still require funding, infrastructure building and education of primary care staff as well as innovative secondary specialists that can build a suitable health care system that will have to be innovative and require much support.

Reply: We agree with the statement, as we have implied in our conclusion section. We have added a sentence in the conclusion section, paragraph 1, sentence 3, to address this issue, as follows:

“Improving the quality of primary care by providing better infrastructure and developing the competence of health personnel may have a large impact on population health considering the (equality in) accessibility of primary care, and could potentially reduce the burden of secondary care.”

Reviewer 3

Reviewer Name: Zhonghua Wang

Institution and Country: Nanjing Medical University, China

Comments to authors:

The reviewed manuscript evaluated socioeconomic inequalities in healthcare utilisation in Indonesia using the data from a national representative survey. Educational inequalities and income inequalities in different types and levels of health care use were measured by the relative index of inequality (RII). The findings have potential to help guide future policy decisions. The manuscript has some general and specific concerns.

We thank the reviewer for the comprehensive review and constructive feedback. We have provided our response to the specific comments (in italic), as follow:

1. Method

a. Page 6. Study design and data sources. Additional information is required about the IFLS5 survey methodology so that the reader would probably benefit from a direct reference to the IFLS5 baseline protocol or methodology paper, to see how the sample was selected and how it is representative of the population.

Reply: To make it clearer to the reader, we have revised the sentence in our manuscript in the methods section, subheading study design and data sources, paragraph 1, sentence 2, as follows:

“The data and supporting documents such as the survey protocol and questionnaires are publicly accessible through RAND’s website.²⁴”

Moreover, in the data sharing statement section, we added:

“This study used the Indonesian Family Life Survey (IFLS) dataset provided by RAND Corp. The IFLS datasets (including the supporting documents such as survey protocol and questionnaire) are freely accessible at <https://www.rand.org/labor/FLS/IFLS.html>.”

b. Page 8 line 24. I recommend that the author gives a more detailed introduction about RII. For example, the rationale for the analysis should be well explained, allowing for the study to be repeated.

Reply: We provided a comprehensive explanation on how to calculate and interpret RII in our method section, subheading data analysis, paragraph 2, sentence 1-4. However, due to the space limit as set by the journal’s requirement, we did not describe in detail the RII calculation. However, we have now added a sentence with reference about RII calculation in the method section, subheading data analysis, paragraph 2, sentence 5-7, as follows:

“Details on how RII calculated can be found elsewhere.²⁸ RII has property to estimate the magnitude of inequalities in one single measure that uses information from all socioeconomic groups individually and allows comparison between different socioeconomic and outcome indicator. RII is commonly used in epidemiological research and has a relatively straightforward interpretation for readers who have no economics background compared to other common inequality measurements such as the concentration index.”

2. Results

In the data analysis, the author mentioned “the rate difference and the rate ratio complement the RII”. Thus, I think the rate difference and the rate ratio were also important ways of describing inequalities

of healthcare utilisation in the manuscript. Additional information is required about the findings of it in the result section, like value and meaning.

Reply: We agree that simple inequality measurements using rate difference and rate ratio are important ways to describe inequality. We decided only to highlight the findings of the sophisticated measurement inequality (RII) in our results section for several reasons. First, the results between simple relative inequality (rate ratio) and sophisticated relative inequality (RII) measurement were similar (no contradictory results). Second, we thought that describing all findings in the tables (with similar results) in the results section narrative will expand the text while it would have little additional value to the readers. We provided supplementary tables and figures in the supplementary file for the readers who have further interest in more detailed results. To make it clearer for the reader, we have added the following sentences in the manuscript:

Results section, paragraph 3, sentence 2:

“Our findings from simple inequality measurement (rate ratio and rate difference) showed similarities with the findings from sophisticated inequality measurement (RII).”

Results section, paragraph 6, sentence 2:

“Our analyses showed consistent findings between simple (rate difference and rate ratio) and sophisticated (RII) inequality estimations.”

3. Discussion

a. In the result section, the author mentioned “larger inequalities were found in relationship to income as compared to educational level, except for utilisation of outpatient secondary care.”. I think the author should further explain the reasons in the discussion section. For example, why the educational inequalities were larger than income inequalities for outpatient secondary care?

a. Reply: We have discussed why educational inequalities were larger than income inequalities for outpatient secondary care in our discussion section, paragraph 8, sentence 3-6. To make it clearer for the readers, we have revised the sentences as follows:

“Inequalities in secondary care may also be influenced by differences between educational groups in the preferences and resources that influence the way people utilise healthcare.³⁵ An Indonesian study showed that patients with higher educational level, regardless of their income level, were more likely to judge the quality of primary care to be low and to ask for a referral to secondary care. This tendency was not observed among people with high income, but relatively low education.^{36 37} Education-related preferences might explain why educational inequalities in outpatient secondary care were larger compared to income-related inequalities.”

b. There is no plan for the next step in this manuscript.

Reply: Following the suggestion, we have added the direction of further research in the conclusion section, paragraph 1, sentence 5-6, as follows:

“Further research is needed to assess inequalities in healthcare among specific patient groups, and to evaluate the contribution of patient preferences and resources, and to examine the role of geographical factors and health care organisation and infrastructure. Such in-depth analyses could provide a better understanding of socioeconomic inequalities in healthcare utilisation in Indonesia and guide the development of strategies to address those inequalities.”

Reviewer: 4

Reviewer Name: Mohammad Habibullah Pulok

Institution and Country: Postdoctoral Research Associate - Nova Scotia Health Authority, Halifax, Canada.

Comments to authors:

Thank you very much for giving me the opportunity to review a nicely articulated paper. The article on socioeconomic inequalities in healthcare utilisation in Indonesia is an interesting and well-designed study. I do not have any major comments that could add something important to the manuscript. The

paper is well-written, and the methodology of the paper is sound and robust. The authors have applied the method very well to the available data.

Minor comments:

I would suggest the authors develop the motivation of the paper a little bit. The introduction would be more attractive if the authors could draw a storyline at the end of the introduction. The motivation is currently statistical which lacks an argument to conduct the study. In this case, the authors could consider adding the current policy perspective in Indonesia to achieve equity.

Reply: Thank for the detailed review and constructive feedback to our manuscript.

- We have added more information to highlight the current policy perspective in Indonesia to achieve equity, as follows:

Introduction section, paragraph 4:

“Current policy to achieve equal access in healthcare in Indonesia is focusing on the expansion of the NHI program.¹⁴ However, over the years, progress towards universal health coverage has been uneven and iterative and consistently driven by domestic political interests as opposed to technical considerations.¹⁵ The dominance of political interest is also reflected in the government monitoring and evaluation of the NHI program which emphasised the overall coverage (NHI membership) of the population and paid less attention to the issue of the actual access distribution such as inequality among various population.¹⁶ “

The authors could improve the discussion section by explaining the possible reasons for lower education-related inequality in some outcomes of healthcare use. The author could also add one or two sentences to outline the future direction of research in the Indonesian context.

- Reply: Following your and previous reviewer’s suggestion, we have revised our conclusion section to address the issue of future research. Please refer to our reply for reviewer 3, point #3b.

- Reply: We have provided more explanation about lower education-related inequalities in primary care compared to secondary care and inpatient care, also for blood pressure measurement compared to other types of preventive care, as follows:

Discussion section, paragraph 5, sentence 4-6:

“In the NHI program, primary care acted as a gatekeeper which required all beneficiaries regardless their socioeconomic background (poor people or government employee) to use primary care as an entry point to access healthcare service.³⁵ For people without insurance coverage, primary care is relatively affordable and can be accessed at low cost, even in private practices.¹³ This likely explained the smaller income and educational-related inequalities in primary care utilisation compared to the inequalities in secondary and inpatient care utilisation.”

Discussion section, paragraph 8, sentence 1-2:

“Inefficient referral procedures could also have contributed to larger inequalities in secondary care utilisation compared to primary care particularly the educational-related inequalities. Even when low-educated people are entitled to access secondary healthcare, they may lack the knowledge required to obtain a referral, due to the complexity of the administrative procedures in the referral system.³⁴ “

Discussion section, paragraph 9, sentence 3-4:

“Those with a relatively low level of health literacy may experience cognitive barriers to make decisions regarding diagnostic tests and treatments that they may need, irrespective of financial, geographic or administrative barriers.^{40 41} This also likely explains relatively smaller educational-related inequalities in blood pressure measurement compared to other types of preventive care because blood pressure disorder such as high blood pressure is relatively known by common people regardless of their educational background compared to other types of preventive care.

I would suggest the authors consider the following recent paper from Indonesia to get some insights to enrich the manuscript.

Johar, M., Soewondo, P., Pujisubekti, R., Satrio, H.K. & Adji, A. 2018, 'Inequality in access to health care, health insurance and the role of supply factors', *Social Science & Medicine*, vol. 213, pp. 134–45

- Reply: We have included the suggested reference in our introduction section and discussion section, as follows:

Introduction section, paragraph 8, sentence 4:

“A recent study showed wealth-related inequalities in Indonesia in the use of health care, particularly in secondary care. However, this study did not assess inequalities in relation to other SES indicators such as educational level, nor did it consider inequalities in preventive care utilisation.²³”

Discussion section, paragraph 4, sentence 3:

“Similar to the recent study on wealth-related inequality in healthcare utilisation in Indonesia, we found smaller inequalities in the utilisation of primary care, especially outpatient care, and larger inequalities in secondary care.²³”

Discussion section, paragraph 5, sentence 4:

“Moreover, according to a recent study, access to primary care was increased by a government-financed NHI program that aimed to reduce financial barriers of the poor population to healthcare.²³”

Discussion section, paragraph 6, sentence 3:

“Because most secondary care facilities and specialists are located in urban areas, the poor need to pay high indirect costs (in terms of travel and opportunity) to access secondary care, even if their medical costs are covered by the NHI program.²³”

VERSION 2 – REVIEW

REVIEWER	zhonghua wang nanjing medical university, China
REVIEW RETURNED	28-Mar-2019

GENERAL COMMENTS	1. Study design and data sources. The author had better summarize the contents of documents in some words so that reader can understand the methodology of the IFLS survey directly, although these documents can be found on websites. 2. In the data analysis, there are some repetitions in the sentence 1-2 and sentence 6 of the second paragraph. 3. I still recommend that the author had better further describe the calculation of RII in the data analysis to ensure the integrity of the paper.
---

VERSION 2 – AUTHOR RESPONSE

Reviewer: 3

Reviewer Name: Zhonghua Wang

Institution and Country: Nanjing Medical University, China

We thank for the valuable comments and constructive feedback. We provide our response to the specific comments (in italic), as follow:

Comments to authors

1. Study design and data sources.

The author had better summarize the contents of documents in some words so that reader can understand the methodology of the IFLS survey directly, although these documents can be found on websites.

Reply:

We added several sentences to provide more explanation about the methodology of IFLS in the method section, subheading study design and data sources, paragraph 1, as follows:

“The IFLS5 is a longitudinal survey which has been conducted since 1993 (IFLS1) and collected data from 13 selected Indonesian provinces to maximally capture the diversity in socioeconomic and cultural background of the Indonesian population. These 13 provinces represented 83% of the Indonesian population. The IFLS used stratified random sampling based on province and rural/urban location. The sampling frame was randomly chosen from the list enumeration area (EA) of the National Socioeconomic Survey which was conducted by the National Bureau of Statistics in more than 60,000 households. Within each urban EA, 20 households were randomly selected while 30 households were selected from each rural EA. In total, 7730 households from 321 EAs in 13 provinces were sampled for IFLS.”

2. In the data analysis, there are some repetitions in the sentence 1-2 and sentence 6 of the second paragraph.

Reply:

Thank you for pointing this out, we have removed the redundant sentences.

3. I still recommend that the author had better further describe the calculation of RII in the data analysis to ensure the integrity of the paper.

Reply:

We have added more explanation about the calculation of RII in the data analysis section, paragraph 2, as follows:

“We assigned a fractional rank of the socioeconomic indicators (income and education) and used these variables as the main predictor in the logistic regression model (considering the binary outcome of outpatient and inpatient care utilisation). The RII was obtained from the value of odds ratio (OR) from the fractional rank of the socioeconomic indicators.”